computational biology, genetics

gene drive, resistance, population control, nuclease deposition

**Author for correspondence:**
Andrea K. Beaghton
e-mail: a.beaghton@imperial.ac.uk

One contribution to a Special Feature 'Natural and synthetic gene drive systems', guest edited by Professor Nina Wedell, Dr Anna Lindholm, Tom Price.

# Gene drive for population genetic control: non-functional resistance and parental effects

Andrea K. Beaghton[1], Andrew Hammond[1], Tony Nolan[2], Andrea Crisanti[1] and Austin Burt[3]

[1]Department of Life Sciences, Imperial College, South Kensington, London SW7 2AZ, UK
[2]Liverpool School of Tropical Medicine, Pembroke Place, Liverpool L3 5QA, UK
[3]Department of Life Sciences, Imperial College, Silwood Park, Ascot, Berkshire SL5 7PY, UK

AKB, 0000-0002-2403-186X; AH, 0000-0002-1757-5009; AB, 0000-0001-5146-9640

Gene drive is a natural process of biased inheritance that, in principle, could be used to control pest and vector populations. As with any form of pest control, attention should be paid to the possibility of resistance evolving. For nuclease-based gene drive aimed at suppressing a population, resistance could arise by changes in the target sequence that maintain function, and various strategies have been proposed to reduce the likelihood that such alleles arise. Even if these strategies are successful, it is almost inevitable that alleles will arise at the target site that are resistant to the drive but do not restore function, and the impact of such sequences on the dynamics of control has been little studied. We use population genetic modelling of a strategy targeting a female fertility gene to demonstrate that such alleles may be expected to accumulate, and thereby reduce the reproductive load on the population, if nuclease expression *per se* causes substantial heterozygote fitness effects or if parental (especially paternal) deposition of nuclease either reduces offspring fitness or affects the genotype of their germline. All these phenomena have been observed in synthetic drive constructs. It will, therefore, be important to allow for non-functional resistance alleles in predicting the dynamics of constructs in cage populations and the impacts of any field release.

## 1. Introduction

Some species—relatively few—cause substantial harm to human health or the environment, and consequently are subject to significant control efforts. A potentially novel way to control such species is to use synthetic nuclease-based gene drive constructs to disrupt one or more genes that are needed for survival or reproduction [1]. Introducing a relatively small number of organisms carrying such a construct into a population could, over time, lead to population-wide knock-out of the target gene, and a consequent reduction in population numbers. There have been a number of mathematical and computer modelling studies showing that such an approach could be effective [2–4], and promising progress in the laboratory demonstrating that it may be feasible to build such constructs, at least in anopheline vectors of malaria [5,6].

As with any form of pest control, due consideration should be given to the possibility that resistance will evolve, and for this approach, the most obvious form of resistance is a change in the sequence recognized by the nuclease such that it is no longer cleaved, but maintains its function in the organism. Such changes may pre-exist in the population, arise by a spontaneous mutation, or be created by the nuclease itself through end-joining repair, and could quickly spread through a population, leading to loss of the driving construct and nullifying the intervention [2,7–13]. Various strategies have been proposed for

reducing the probability of such target-site resistance evolving, including targeting sites that are functionally constrained and targeting multiple sites in the same gene, and there has been some progress with both approaches [6,11,14,15]. However, even if functional target-site resistance is completely avoided, it is almost inevitable that non-functional resistant alleles at the target site will arise, at least at some frequency. These are sequences that are not recognized and cleaved by the nuclease and do not restore survival or reproduction for the organism, and may arise by a spontaneous mutation, end-joining repair and imperfect homologous repair. The impact of these sequences on the dynamics of a driving construct and the extent of population suppression has been less well studied.

In order for a driving construct that disrupts a gene needed for survival or reproduction to spread in a population, it must not impose a large fitness cost on the heterozygote. The target gene and control sequences used to drive expression of the nuclease have to be chosen accordingly [2]. For example, the target gene may only be needed in somatic cells, while the nuclease is expressed only in the germline. However, in the first attempts to engineer this type of suppression drive, the control sequences showed some unintended (leaky) expression in somatic cells, resulting in some of those cells (where the target gene is needed) being homozygous null, thereby reducing fitness [5]. Importantly, because this fitness cost is due to expression of the nuclease, it is not borne by non-functional resistant genes, which can, therefore, have a relative advantage and accumulate to higher frequencies in the population than they otherwise would.

There have also been repeated observations in the laboratory of parental effects, whereby the fitness of an individual depends not only on its zygotic genotype, but also on the parental genotypes, apparently due to deposition of nuclease RNA and/or protein into the gametes, followed by cleavage of the target site in some fraction of the embryonic cells [5–7,9,11–13,16–19]. Depending on the construct, both maternal and paternal deposition have been observed. Such deposition has also been observed to affect the offspring's germline and the genetic composition of the gametes it produces. These parental effects may be expected to act analogously to leaky expression in allowing non-functional resistant alleles to accumulate to higher frequencies than otherwise, reducing the extent of population control.

In this paper, we use population genetic modelling to investigate the quantitative impact of these laboratory-observed phenomena on the spread and impact of driving constructs. The specific scenario is a construct targeting a female fertility gene, similar to those that have been reported in *Anopheles gambiae* mosquitoes [5,6]. If, as has been observed, nuclease expression causes strong fitness effects in drive heterozygotes, or if parental deposition of nuclease reduces offspring fitness or affects the genotype of their germline, we find that there can be reductions in the equilibrium frequency of the driving construct and the reproductive load imposed upon the population. Under such conditions, non-functional drive-resistant alleles can accumulate in the population and reduce the extent of population control.

## 2. Models

We model a random-mating population with discrete generations, two sexes, and three alleles: wild-type (W), driver (D) and resistant (R). D alleles encode a nuclease that can cleave W alleles, converting them to D or R alleles depending on the type of repair. At the sequence level, there may be many different R alleles, but we assume they all behave the same: they are not cleaved by the nuclease, and they are non-functional with respect to female fertility. Functional resistance is assumed not to be possible. As the W allele is needed for female fertility, D/D, D/R and R/R females are sterile. The fitness of W/D and W/R females are allowed to vary, and we assume no fitness effects on males.

Leaky expression of the nuclease in W/D females may lead to individuals that are mosaic in their soma, with a proportion of cells R/D due to end-joining repair (and D/D for homologous repair). We model this by a reduction in fitness of W/D heterozygote females. To model the action of deposited parental nuclease in the embryo, we assume that individuals may end up mosaic in their soma, affecting female fitness, and/or mosaic in their germline cells which alters gene transmission (figure 1). For the effect of parental deposition on offspring fitness, we follow Kyrou *et al.* [6]. In brief, if the zygotic genotype contains at least one W allele, and at least one of the parents contained a D allele, then deposited nuclease may act on the W allele and reduce fitness of females to $w^{10}$, $w^{01}$ or $w^{11}$ depending on whether the nuclease was derived from a transgenic mother, father or both, respectively. We assume that parental effects are the same whether the parent(s) had one or two D alleles. To model the effect of parental deposition on germline cells and gamete production in the offspring, we assume that cleavage of the W allele by the deposited nuclease results in a fraction $\delta_e^a$ (embryonic EJ rate) of germline stem cells in the embryo undergoing cleavage and end-joining repair (producing an R allele), and a fraction $e_e^a$ (embryonic HR rate) undergoing homologous repair, producing R or D alleles (depending on the homologous allele), where the superscript $a$ is 10, 01 or 11 to denote deposition from mother, father or both, respectively ($\delta_e^a + e_e^a \leq 1$ for all $a$). For simplicity, we assume these effects are the same in male and female offspring. (It would be straightforward to incorporate such differences, if, for example, the cleavage occurred after sexual differentiation, or the W allele has a sex-specific role in one of the germlines.) The resulting proportions of germline stem cells are shown in electronic supplementary material, table S1. There are five types of sperm and eggs in this model (W and R without deposited nuclease and W*, D* and R* with nuclease), and 14 types of females and males, and the proportions of gametes produced by each zygotic type are shown in electronic supplementary material, table S2.

These assumptions are incorporated into a system of difference equations that allow one to calculate the frequency of the different genotypes in one generation as a function of their frequency in the previous generation. We use these equations to track the frequencies of the alternative alleles over time, and the reproductive load imposed on the population, defined as the proportionate reduction in reproductive output by the population (see electronic supplementary material, Model Description). All simulations start with an initial frequency of 1% W/D among the male population, with all other males and females being W/W, and equilibrium allele frequencies and load are taken after 200 generations. We investigate variation in embryonic rates of end-joining and homologous repair ($\delta_e^a$ and $e_e^a$), with baseline values for those at meiosis ($e_m$ and $\delta_m$) of 90% and 5%, respectively,

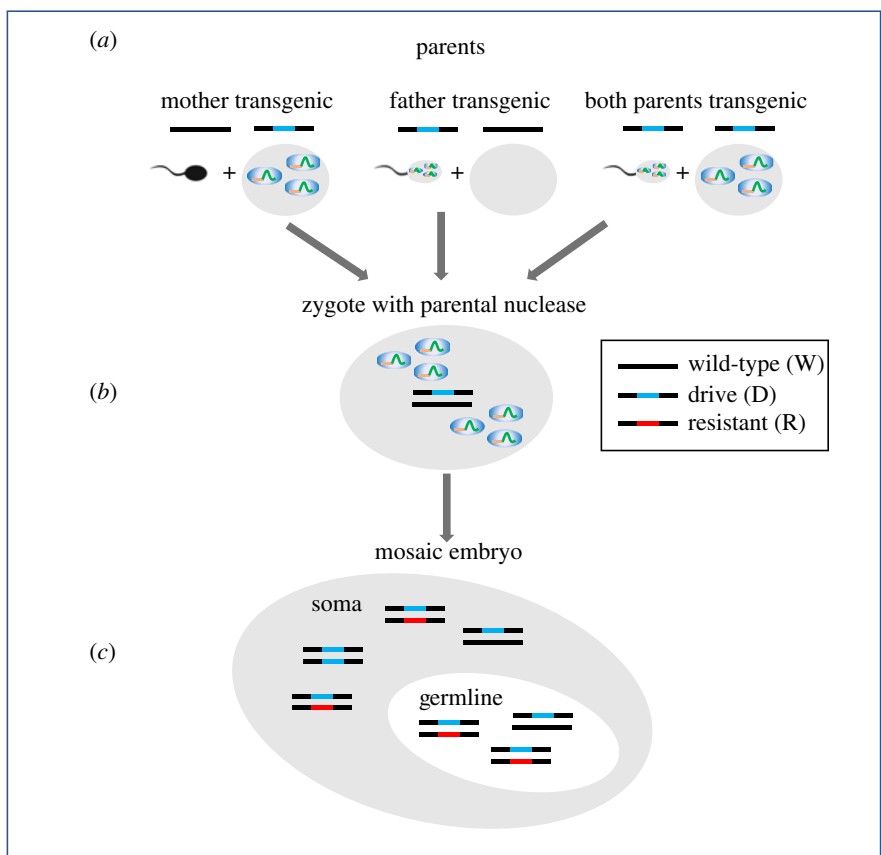

**Figure 1.** Model of parental deposition of nuclease, shown here for a W/D zygote, although we assume that W/R and W/W zygotes can also have cleavage due to deposited parental nuclease. (*a*) Deposition of the nuclease into the egg from a transgenic mother, into the sperm from a transgenic father or into both egg and sperm from both transgenic parents (note that a gamete carrying a wild-type (or resistant) allele from a transgenic parent may also carry deposited nuclease). (*b*) Deposited nuclease in the zygote may cause cleavage of the wild-type allele and repair (by end-joining or homologous recombination) and can happen at any time from the formation of the zygote until the stage where germline stem cells are formed. (*c*) Once the germline stem cells are formed, the model assumes that the nuclease is no longer active. The individual may then be mosaic in the soma (effects on fitness) and/or in the germline (effects on gene transmission). Gene transmission from the germline cells is then according to the progeny's own nuclease.

comparable to values observed in the laboratory in *An. gambiae* [5,20]. Parameter definitions and baseline parameter values are given in electronic supplementary material, table S3.

## 3. Results

### (a) Leaky expression

To investigate the effect of leaky expression, we compare the dynamics of allele frequencies and load (i.e. the reduction in reproductive output by the population) under three scenarios: first, when there are no heterozygous fitness effects of either the D or R alleles; second, when they both show heterozygous fitness reductions of 60% (due to partial haplo-insufficiency of the target gene) and third, when only the D allele shows the fitness reduction (due to leaky expression; figure 2). In the first two scenarios, R alleles arise but remain relatively rare (ca. 5%). The main outcome of the heterozygote fitness effect due to partial haplo-insufficiency is to slow down the spread of the transgene and to increase the eventual equilibrium load. By contrast, with leaky expression, the R allele reaches substantially higher frequencies (greater than 20%), while the frequency of the D allele and the load overshoot their equilibrium values before falling back.

These differences between haplo-insufficiency and leaky expression apply more generally. In the case of haplo-insufficiency, increasing the cost to heterozygotes has little impact upon equilibrium allele frequencies and load on the

population, until the cost is sufficiently high that it counteracts the drive and both the D and R alleles are lost, leaving only W (figure 3*a*). Up until that point, the equilibrium load actually increases with an increasing heterozygous fitness effect. By contrast, with leaky expression, the equilibrium frequency of the D allele and reproductive load both decline continuously with increasing cost (figure 3*b*). With haplo-insufficiency, the R allele has the same fitness effect as the D allele, but without the drive, and the only factor maintaining R in the population is a recurrent mutation. As a result, the equilibrium frequency of R is approximately a linear function of the rate at which it originates (figure 3*c*). Again, leaky expression is different: the R allele does not suffer the same fitness cost as D, and is still resistant to being cleaved, and so R can be positively selected. As a result, even vanishingly small (but nonzero) rates of origin lead to a relatively high equilibrium frequency (greater than 15% with our baseline parameter values; figure 3*d*). As a result, when leaky expression does occur, there will be a limit to how much one can ameliorate its effects by trying to reduce the rate of end-joining repair. For both cases, the time taken for the load to reach its equilibrium value increases with higher fitness cost (electronic supplementary material, figure S1).

### (b) Parental effects on fitness

It has been observed experimentally that the fitness of individuals can depend not only on their zygotic genotype,

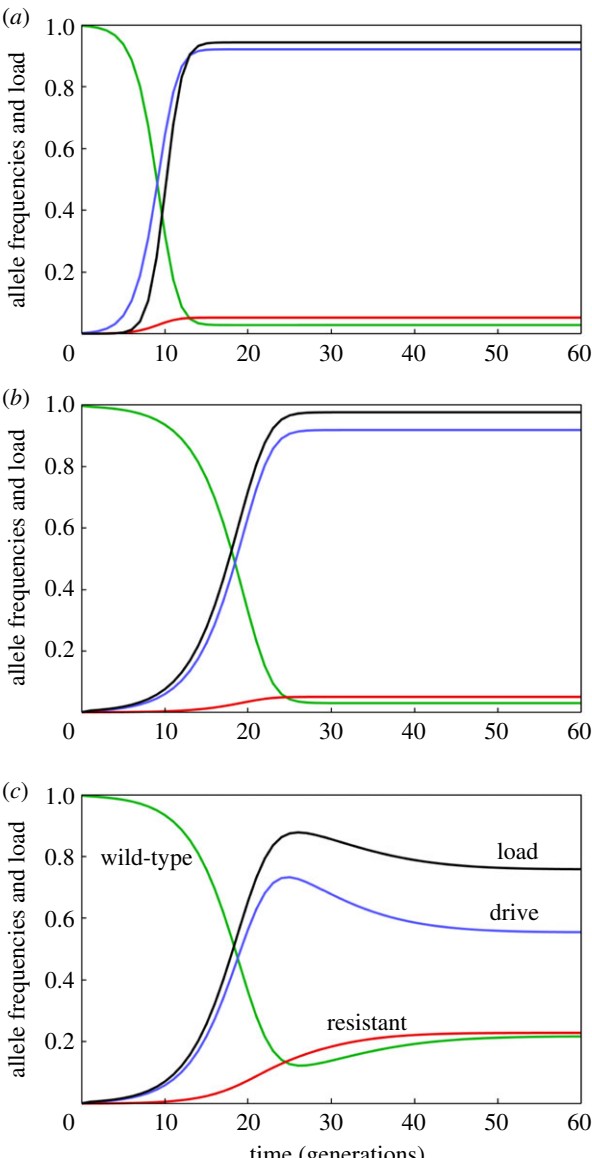

**Figure 2.** Dynamics of a homing construct targeting a female fertility gene, showing the change in load and in the frequency of wild-type, drive and resistant alleles over discrete generations. (*a*) No heterozygous fitness effects (fitness W/D = W/R = 1). (*b*) Partial haplo-insufficiency: fitness W/D = W/R = 0.4. (*c*) Leaky expression (fitness W/D = 0.4, W/R = 1).

but also on the genotype of their parents, apparently due to parentally deposited nuclease creating mosaic offspring. Here, we investigate the consequences of fitness reductions due to parentally deposited nuclease for a strategy targeting a recessive female fertility gene, assuming there is neither partial haplo-insufficiency nor leaky expression. We investigate the sensitivity of the outcome (i.e. the allele frequency and load) to whether the fitness reductions in the offspring are caused by deposition from the mother only, the father only or both, by varying the fitness parameters for each separately. We make no assumptions about whether maternal or paternal deposition is more likely—both have been observed for different constructs—and instead simply investigate the impact of deposition by either parent or both on the outcome. We further assume, at least initially, that deposited nuclease affects the fitness of W/W, W/D and W/R females equally.

If nuclease deposition only occurs from the mother, then there is little effect on the equilibrium frequency of the D

allele or the reproductive load, even if that deposition is lethal to the daughters (figure 4). Indeed, maternal deposition can give a slight increase in the load. By contrast, if the deposition is only from the father, and the effect is large (greater than 80% fitness cost with our baseline parameter values), then both the equilibrium frequency of D and the load can be substantially reduced, and the D allele disappears if the effect is fully sterile. The larger impact of paternal compared to maternal deposition in a strategy targeting a female fertility gene is because D/D and D/R genotypes are fertile if male and sterile if female, and therefore more zygotes have a D-bearing father than D-bearing mother. Finally, if the deposition is from both parents, it makes little difference whether the fitness effect is the same as deposition from one parent or whether the effects combine multiplicatively: in both cases, the impact on the equilibrium frequency of D and load is larger than with deposition from only one parent. Thus, parental deposition effects on offspring fitness can have the same qualitative impact as leaky expression, but the fitness effects need to be stronger to have a significant impact. This is because parental deposition also reduces the fitness of W/R heterozygotes, and therefore the advantage of R alleles relative to D is less than with leaky expression.

As noted, these results are based on the assumption that deposition affects the fitness of W/W, W/D and W/R females equally. It is conceivable that W/W females might be less affected because they have two copies of the target gene and both must be disrupted to produce the fitness effect. However, there is no qualitative (and insignificant quantitative) change to the results if we instead assume that only W/D and W/R females are affected by the deposition, because W/W females with a drive parent are relatively rare.

## (c) Parental effects on gene transmission

Parentally deposited nuclease may not only cleave W alleles in somatic cells, affecting the fitness of the offspring, but also act in germline cells, affecting the genetic composition of the gametes produced by the offspring. Again, this effect could occur in W/W, W/D and W/R individuals, but now males, as well as females, can be affected. The consequences of deposition-associated embryonic germline cleavage will depend on how the cut site is repaired, and in particular whether by homologous recombination or end-joining. In flies and mosquitoes, it appears that end-joining is the dominant form of repair in the early embryo [7,20]. Here, we consider two scenarios: first, that all embryonic repair is by end-joining and second, that the ratio of homologous to end-joining repair is the same as we assume in the gonads (i.e. 18 : 1).

If all deposition-associated cleavage is repaired by end-joining, then increasing the rate of such cleavage leads to a reduction in the equilibrium frequency of the D allele and the reproductive load (figure 5*a,b*). Increased rates of cleavage lead to increased accumulation of R alleles, at least until the D allele frequency decreases sufficiently that individuals with transgenic parents become rare in the population. Then R alleles are rarely formed, and their frequency drops as well as that of D (for paternal and biparental deposition, red dashed lines in figure 5*a*). The accumulation of the R allele is not due to any difference in the fitness effect of D and R, but simply due to the increased rate at which R alleles are formed. Again, paternal deposition has a larger effect than maternal, and deposition from both parents a larger effect

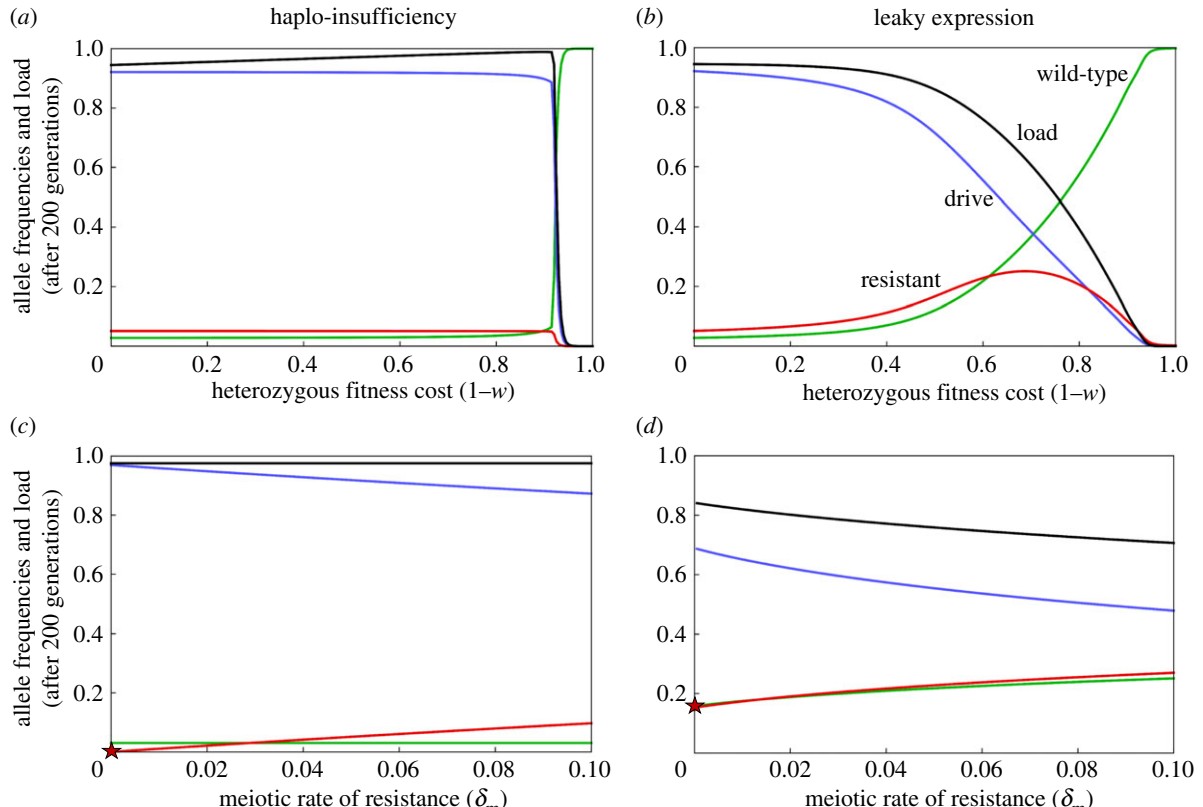

**Figure 3.** Equilibrium allele frequencies and load as a function of heterozygote fitness cost ($a,b$; a value of 1 indicates complete sterility) or the probability R alleles are formed ($c,d$) in the case of partial haplo-insufficiency ($a,c$) or leaky expression ($b,d$). In ($c,d$) the star indicates the equilibrium frequency of R when the population is seeded with 0.01% R alleles, but otherwise they do not arise during the simulations ($\delta_m = 0, \delta_e = 0$). In ($c$) female fitness W/D = W/R = 0.4 and in ($d$) female fitnesses are W/D = 0.4, W/R = 1.

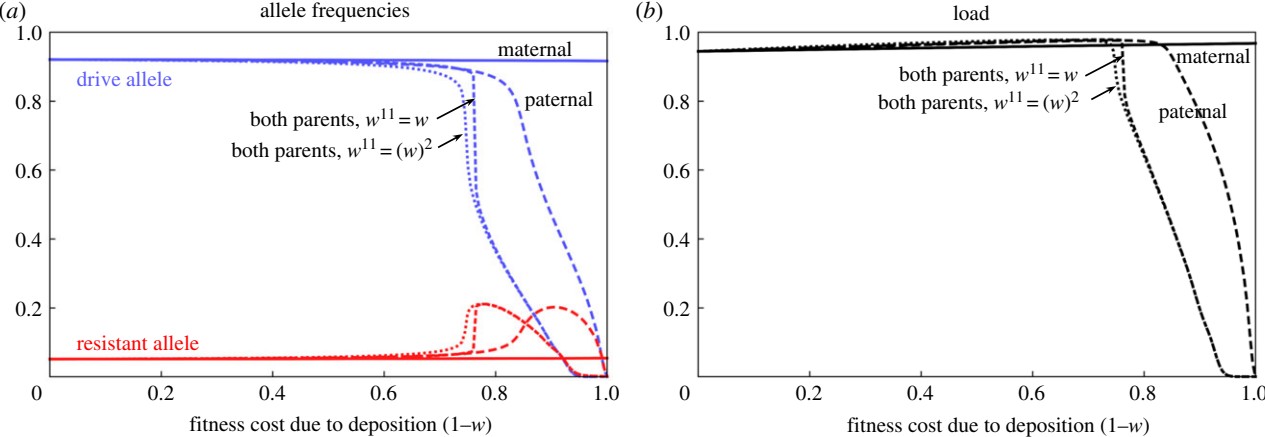

**Figure 4.** Equilibrium frequency of D (blue lines) and R (red lines) alleles ($a$) and load ($b$) as a function of the reduction in W/W, W/D and W/R female fitness due to parental deposition. For maternal deposition only ($w^{10} = w$, $w^{01} = 1$, $w^{11} = w$, solid lines), paternal deposition only ($w^{10} = 1$, $w^{01} = w$, $w^{11} = w$, large dashed lines) and deposition from both parents ($w^{10} = w^{01} = w^{11} = w$; medium dashed lines or $w^{10} = w^{01} = w$ and $w^{11} = (w)^2$; dotted lines).

than from either one alone. For example, for our baseline parameters, if deposition always leads to cleavage and end-joining repair ($\delta_e = 1$), then the equilibrium load is 0.80 for maternal deposition, 0.30 for paternal and $4 \times 10^{-4}$ for biparental (the construct is virtually lost), compared to a load of 0.945 if there is no deposition (figure 5$b$).

As expected, the results change dramatically if embryonic cleavage is predominantly followed by homologous repair (figure 5$c,d$). Under this scenario, maternal deposition has little effect on equilibrium allele frequencies or load, while paternal deposition leads to a decrease in the frequency of the D allele, with an increase in the frequency of R,

presumably because R alleles are additionally propagated by embryonic homing. Interestingly, these changes in allele frequency are associated with a modest *increase* in the (already high) equilibrium load, because the frequency of the W allele (upon which the fertility of the population depends) decreases.

## (d) Combined parental effects on fitness and gene transmission

Parental deposition can obviously affect both fitness and gene transmission, and while a full analysis of the joint effect is

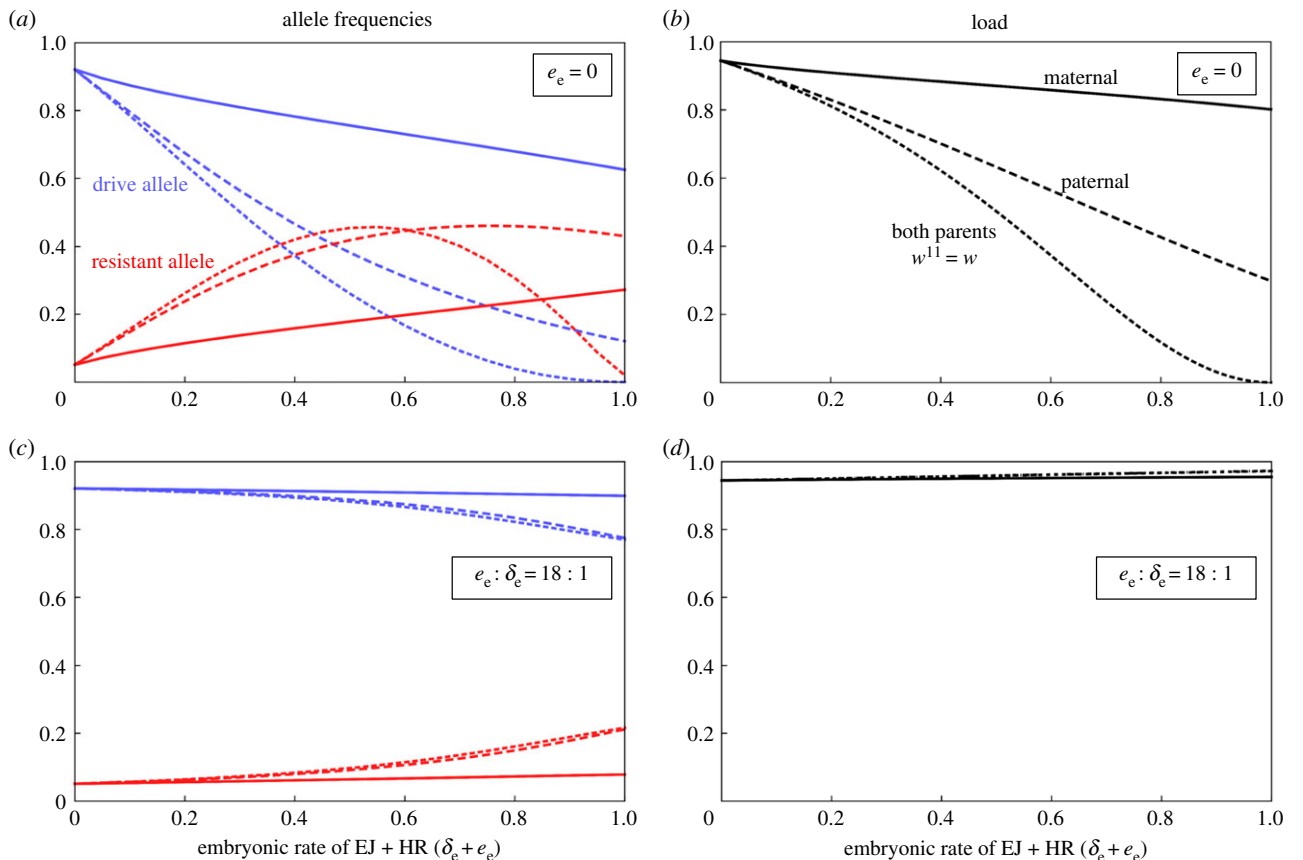

**Figure 5.** Equilibrium D (blue lines) and R (red lines) allele frequency (a,c) and load (b,d) as a function of the embryonic cleavage rate from parental deposition when all repair is by end-joining (a,b) or when the ratio of homologous to end-joining repair is $e_e : \delta_e = 18 : 1$ (i.e. the same as in the gonads). In (a,b), where $e_e = 0$, deposition is maternal only ($\delta_e^{10} = \delta_e$, $\delta_e^{01} = 1$, $\delta_e^{11} = \delta_e$, solid lines), paternal only ($\delta_e^{10} = 1$, $\delta_e^{01} = \delta_e$, $\delta_e^{11} = \delta_e$, large dashed lines) or biparental assuming $\delta_e^{11} = \delta_e$ (medium dashed lines). In (c,d) deposition is maternal only (solid lines), paternal only (large dashed lines) or biparental (medium dashed lines).

beyond the scope of this paper, representative results (for the case of embryonic cleavage always being repaired by end-joining) are shown in electronic supplementary material, figure S2. As expected, increasing frequencies of deposition-associated embryonic cleavage lead to reductions in the equilibrium frequency of the D allele and the equilibrium load in this case too.

# 4. Discussion

The possibility that a nuclease-based gene drive might select for functional target-site resistance is well acknowledged, and various possibilities are being explored to reduce the likelihood of this occurring [6,11,14,15]. The potential for non-functional target-site resistance to be selected for and reduce the efficacy of control has been less well discussed. Previously, it has been shown in the context of a driving construct targeting an essential gene that if both D and R alleles cause recessive lethality in both sexes then their equilibrium frequencies (using our current notation) are $e_m^2/(e_m + \delta_m)$ and $e_m \delta_m/(e_m + \delta_m)$, respectively, and the equilibrium load is $e_m^2$ [2]. With the parameter values typically observed in the laboratory, at least in *An. gambiae*, non-functional resistance would be expected to remain rare. However, these same laboratory experiments have also revealed some factors not taken into account in the simple model, including leaky expression and parental deposition of the nuclease. As we have demonstrated, these factors can be expected to increase the frequency and importance of non-functional resistance.

Neither leaky expression nor parental deposition are inherent features of a driving construct and could be ameliorated by appropriate design. Both may be altered by changing the control sequences driving expression of the nuclease [20], and parental deposition, at least, by changing the half-life of the nuclease [17]. Further strategies for controlling nuclease expression, such as synthetic regulatory circuits or quenchers, have also been proposed [21]. Note that in some other contexts, such as designing some self-limiting genetic control methods, it may be desirable to enhance parental deposition [22].

Our model of parental deposition extends that of Kyrou *et al.* [6] to include effects on the offspring germline. It explicitly allows for the somatic and germline mosaicism observed in the laboratory [6,7,11–13,15,18,19], and therefore better captures the underlying biology than simple models that only allow zygotic action of deposited nuclease and do not produce mosaics [9,23]. The trade-off is an increase in model complexity, from six to 14 different types of males and females. Despite the large number of genotypes, we have tried to keep the number of parameters small. We assumed that the effect of parental deposition on fitness of W/W and W/R offspring was the same as for W/D, reducing nine possible mosaic fitnesses to two parameters ($w^{10}$ and $w^{01}$, with $w^{11}$ a function of these two). Similarly, for the transmission effects of deposition, we simplified 18 possible parameters (36 if the two sexes were treated separately) to just four ($\delta_e^{10}$, $\delta_e^{01}$, $e_e^{10}$ and $e_e^{01}$).

Effects on fitness are often estimated from the rate at which a gene increases or decreases in frequency in multi-generation population cage experiments [9]. For genes that

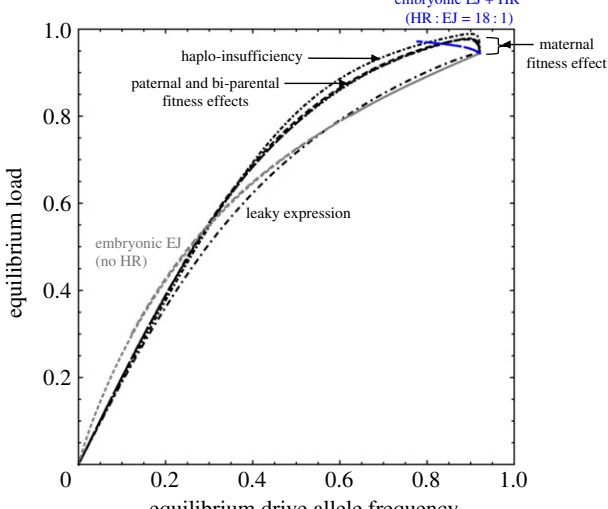

**Figure 6.** Equilibrium load (after 200 generations) as a function of equilibrium frequency of the D allele, with different lines corresponding to underlying variation (from 0 to 1) in different parameters as follows. Black lines denote varying female heterozygote fitness due to: somatic leakage (large dot-dashed line), haplo-insufficiency (small dot-dashed line), maternal deposition (short solid line indicated by brackets), paternal deposition (large dashed line) or biparental deposition when $w^{10} = w^{01} = w^{11} = w$ (medium dashed line) and when $w^{10} = w^{01} = w$ and $w^{11} = (w)^2$ (dotted line). Grey lines show the results of varying the germline embryonic cleavage rate when all repair is by end-joining ($e_e = 0$) with effects from mother, father and both (solid, large dashed and medium dashed lines, respectively—note they all fall on the same line), with no effect on fitness. Blue lines similarly denote the results of varying the embryonic cleavage rate, except the ratio of homologous to end-joining repair is $e_e : \delta_e = 18 : 1$.

tend to an intermediate equilibrium frequency, measurements of that frequency can also be used to estimate fitnesses [24,25]. To determine whether a similar approach might be used to estimate the equilibrium load, we varied all the factors considered in this paper and plotted their joint effect on the frequency of the transgene and the load (figure 6). As can be seen, the relationship between these two parameters is expected to fall in a relatively narrow band regardless of the underlying cause of the variation, whether

it be haplo-insufficiency, leaky expression or deposition effects on fitness or transmission. Note, however, it is still the case that the success of the prediction will depend on the extent to which fitness effects in the cage are the same as fitness effects in the field. As long as these effects are either very small or complete sterility or lethality, this extrapolation may be straightforward, but for the more intermediate effects included here, it will be particularly important for the cage environment to mimic as closely as feasible the relevant aspects of the natural environment.

Our modelling could in future be extended in several directions. First, we have taken the equilibrium to be the state reached 200 generations after a release of 1% W/D males, but it is possible that other equilibria exist that would be reached from different initial conditions. Second, we have used reproductive load as a proxy for the efficacy of control. In simple non-spatial models of density-dependent regulation, if homozygous mutant females are both sterile and unable to transmit disease, then the proportionate reduction in disease transmission is a linear function of the load (up to a maximum of 100%; eqn S5a′ in Deredec *et al.* [3]). However, in spatial models with the potential for re-colonization of previously cleared areas, the relationship between protection and load can be more complex [8,26]. In particular, it is possible that protection may be maximized at an intermediate optimum load and then decline as load increases beyond that. The relationship between load and protection needs further study. Finally, it is worth noting that the factors investigated in this paper—leaky expression and parental deposition—tend to reduce the frequency of the transgene, and therefore one might expect that if mutations arise in the construct to ameliorate these effects, those mutations would be selected for. Similarly, because these factors tend to reduce fitness, there may also be selection acting elsewhere in the genome to reduce them. Further modelling would be needed to test these ideas and determine the timescale over which such selective substitutions could occur.

Data accessibility. This article has no additional data.

Competing interests. We declare we have no competing interests.

Funding. This work was supported by grants from the Bill & Melinda Gates Foundation and the Open Philanthropy Project.

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
