## [Reviewer comments · Proceedings of the Royal Society B: Biological Sciences]

Review History

RSPB-2019-1586.R0 (Original submission)

Review form: Reviewer 1

Recommendation

Accept with minor revision (please list in comments)

Scientific importance: Is the manuscript an original and important contribution to its field?

Good

General interest: Is the paper of sufficient general interest?

Good

Quality of the paper: Is the overall quality of the paper suitable?

Good

Is the length of the paper justified?

Yes

Should the paper be seen by a specialist statistical reviewer?

No

Do you have any concerns about statistical analyses in this paper? If so, please specify them explicitly in your report.

No

It is a condition of publication that authors make their supporting data, code and materials available - either as supplementary material or hosted in an external repository. Please rate, if applicable, the supporting data on the following criteria.

Is it accessible?

Yes

Is it clear?

Yes

Is it adequate?

Yes

Do you have any ethical concerns with this paper?

No

Comments to the Author

In the manuscript, Beaghton and colleagues take the next step in studying the evolution of resistance to gene drive by exploring the influence of parental deposition and leaky expression of gene drive nucleases. They find that both can influence the equilibrium values of both driver frequency and drive load. The paper is scientifically sound and mostly well written. I have several comments:

Lines 86-91 - I could use more of a preview in this paragraph. Instead of referring to "these factors", the authors could rehash what the factors are and give a bit more of a hint at the answer. It took me a while to understand exactly what the paper was about.

Line 107 (and throughout) - I struggled with mosaicism in the manuscript in general. It seems like leakiness and parental deposition would very likely lead to mosaic animals where fitness would be depend (linearly, based on a threshold, etc.) on the proportion of cells that are WW/WD/WR/DD/DR/RR after editing. Perhaps the assumption is that all editing takes place early enough that mosaics are unlikely, but the discussion of somatic mutations would argue against that. Perhaps I'm just unclear on the assumption, but some clarification would help.

Line 107/114 - the use of the superscript 01/10/11 in two different contexts is confusing. Can one be changed?

Line 129 - how certain are you that for all parameter space, equilibrium is actually reached in 200 generations - that seems awfully short

"Parental effects on fitness" section - I had trouble following this section. It would benefit from some clarification about the question being asked and how it is addressed.

I would really like an intuitive explanation for why paternal deposition had a much stronger influence than maternal effects. It seems like they should be symmetrical. Can the authors provide any insight?

Lines 219-220 - The authors state that "However, there is no qualitative (and very little quantitative) change to the results if we instead assume that only W/D and W/R females are affected by the deposition." Is this worth putting in a supplemental table?

Mathematica - It might be useful to provide the Mathematica notebooks as a supplement or Figshare.

Review form: Reviewer 2

Recommendation

Accept with minor revision (please list in comments)

Scientific importance: Is the manuscript an original and important contribution to its field?

Good

General interest: Is the paper of sufficient general interest?

Acceptable

Quality of the paper: Is the overall quality of the paper suitable?

Good

Is the length of the paper justified?

Yes

Should the paper be seen by a specialist statistical reviewer?

No

Do you have any concerns about statistical analyses in this paper? If so, please specify them explicitly in your report.

No

It is a condition of publication that authors make their supporting data, code and materials available - either as supplementary material or hosted in an external repository. Please rate, if applicable, the supporting data on the following criteria.

Is it accessible?

No

Is it clear?

Yes

Is it adequate?

Yes

Do you have any ethical concerns with this paper?

No

Comments to the Author

In this manuscript, the authors analyzed homing drives for population suppression using computational modeling. They specifically examined the effect of resistance allele formation in

which the resistance alleles disrupt the target gene of the suppression drive (including alleles formed by parental deposition and in somatic cells by leaky nuclease expression). These alleles won't stop the drive like those that change the sequence but do not disrupt the target gene, but they can still have substantial effects on the outcome of the drive strategy. In particular, the authors find that such alleles can reduce the genetic load on the population, potentially preventing complete suppression of the population and allowing it to persist at a lower equilibrium population.

This is not a fundamentally new concept. Such resistance alleles have been included in a few previous models. Furthermore, in Deredec et al. 2008 and in a recent study posted to bioRxiv (<https://www.biorxiv.org/content/10.1101/679902v1>), it was even shown that complete population suppression would not occur for less efficient drives due to these same factors that the authors discuss. Nevertheless, the authors' treatment of this topic is quite focused (unlike the bioRxiv study, which focuses on other matters and only briefly touches on the main points of this manuscript). It includes explicit treatment of leaky expression and parental deposition, and these topics are certainly an important enough that they deserve a thorough treatment. Indeed, the manuscript is particularly timely in light of the recent experimental publications of successful homing drives for population suppression. It is of overall high quality and should be quite suitable for publication with some modest revisions detailed below (sorry for the length - I get enthusiastic about gene drive). The only serious point is #17.

1. In the abstract, "non-functional" resistance alleles is used without much background. This could be very confusing to people outside the field of gene drive. The authors should consider describing how population suppression drives have a genetic target, and that non-function resistance alleles refer to the target gene. Same with paragraph two in the introduction section.
2. Later in the abstract, it is unclear why paternal nuclease deposition may be more serious than maternal deposition, particularly since maternal deposition would be expected to be far more common, all else being equal.
3. Page 2, line 55: The Oberhofer PNAS reference is not very good for noting progress with multiple gRNAs (also in the discussion section page 10 line 275). Their construct had far lower efficiency than one-gRNA drives in *Drosophila*. The other PNAS paper by Champer et al. in 2018 showed an improvement and should probably be reference here if the authors want to show progress with multiple gRNAs (not necessarily recommending a replacement of Oberhofer, just an addition). Perhaps also mention improved promoters for improving drive efficiency, such as Hammond 2018 on bioRxiv?
4. Page 3, line 68: "Leaky expression" was also observed in some of the *Drosophila* studies. It was noted in the Champer 2018 paper with the *vasa* promoter (in contrast to *nanos*, which didn't have noticeable leaky expression) and in some of the Gantz/Bier papers with the *vasa* promoter.
5. Page 3, line 104: It's a decent approximation to model parental effects to be the same with one or two drive alleles, but I don't think it's a good idea to model males as having a parental effect with the CRISPR system, since there is no hard evidence of this that I'm aware of. The authors are very aware of male parental deposition due to their previous experience with I-PpoI X-shredders, but a similar CRISPR X-shredder (Galizi et al., 2016) that they constructed did not show this phenomenon. Since the female gamete is much larger, it makes sense in general to assume that maternal deposition would usually have a much greater impact.

The reduced fertility they saw in the daughters with males of their homing suppression gene drives in *Anopheles* could probably be better explained by leaky expression of Cas9 rather than male deposition. This is quite clear in the *vasa* drive and the *nanos* drive from bioRxiv where the

drive conversion efficiency of progeny with a drive mother is reduced. The lack of reduced drive conversion in the zpg lines indicates that reduced fitness is more likely due to leaky somatic expression. If there was substantial maternal or paternal deposition, then at least some resistance alleles would be formed in the early embryo and prevent drive conversion, which we do not see to a significant degree. The difference in egg count in females that had a drive mother or father could probably be better explained by imprinting or another phenomenon affecting the degree of leaky expression (especially since it seems different for the doublesex locus and the AGAP007280 gene in their bioRxiv paper).

While I believe that implementing this change would result in improvement in the accuracy/realism of the modeling, the ultimate qualitative outcomes would likely be similar whether maternal or both paternal and maternal nuclease deposition was modeled. Thus, if it would not be possible to easily adjust the modeling, it should be sufficient to briefly note some of these considerations in the discussion of the manuscript.

6. Page 3 line 107 through end of paragraph: I'm a little unclear on one aspect of the model here. If there is parental nuclease deposition, it seems that two things both happen. A) W alleles may be converted to R or D alleles (usually R). B) All females with a drive parent will have a reduction in fitness, depending on which parents have drive alleles.

This seems to be pretty close to my understanding of how things would work, but it could perhaps get some clarification, since this section is pretty dense. Based on my understanding, maternal deposition would form resistance alleles (with no drive conversion) in a fraction of early embryos. This often makes the entire organism D/R or R/R, resulting in complete sterility for females. It would also prevent any drive conversion at all in the germline of males. Then, if there is not cleavage in the early embryo, there could be cleavage later in the embryo that results in a mosaic pattern of cleavage (seen in the *Drosophila* experiments and closely related to the early embryo cleavage rate - see supplemental material in Champer et al. 2019b). This could reduce the fitness of females by a varying amount and possibly affect drive conversion efficiency in the germline if this tissue is affected (this stage only seems to be what the authors model, rather than the "full" embryo resistance alleles formed before formation of mosaicism would take place). Up to this point, these effects could occur regardless of whether an individual actually received a drive allele. Finally, there is leaky somatic expression for individuals with a drive, which can occur whether the drive was received from a male or female parent. This will reduce the fitness of females, but probably not affect germline drive conversion efficiency. As in #5, the authors probably don't need to adjust their model if it is a good approximation, but they may want to note any differences in the above methods in their methods section if they agree with the above description of the mechanism. Either way, please try to improve the clarity of this section. The supplemental section helped, so one possible option is to reduce detail and reply more on the supplement.

7. Page 4, line 130-131. Using "e" for HDR (when e is the first letter in EJ) keeps throwing me off in this manuscript. Part of this I traced to these lines. I suggest flipping the order of and e in the parenthesis that is part of the phrase "We investigate variation in embryonic rates of homologous and end-joining repair (and e)". You could also flip the order before the parenthesis.

8. Figure 2 results: it might be interesting to point out earlier (it is discussed in Figure 3) that while the drive takes longer in the haploinsufficiency scenario, the final load is higher. This could actually be quite important (a 0.99 load is quite different from 0.95 in many scenarios). In this section, it may also be good to redefine load, which is current in the methods (which readers often skip).

9. Figure 3: the clarity of this figure would be increased if the vertical axis was changed from “allele frequencies and load” to “allele frequencies and load after 200 generations”.
10. Figure 3: in (C) and (D), the rate of meiotic resistance is varied from the default value of 0.05. Since this plus the rate of HDR cannot exceed 1, and the default rate of HDR is 0.9, how are the two values related to each other? The rate of HDR would need to be lower than 0.9 if the rate of EJ repair is greater than 0.1 unless these events occur sequentially, but I didn't see any reference to this.
11. Figure 3: as the fitness costs and resistance rate increases, the rate at which the drive reaches a high load will be reduced. A good supplemental figure may show something relating to this (time to 90% of maximum genetic load?) if it could be done quickly/easily.
12. Page 8, line 219: the authors could explicitly say that W/W females with a drive parent would be very rare, which is why changing things for them has little effect on the overall outcome (even though they would undoubtedly be less affected).
13. Figure 4: Since the authors already do not distinguish between parents having one or two drive alleles, it's probably not necessary to separately model $w_{11} = w$ and $w_{11} = w^2$. Just keeping the former is fine with a note in the main text, and this could help make the graph a bit less crowded. This is low priority, though.
14. Figure 4: the vertical axis of this figure has both frequencies and load, but the graphs are one or the other. It may be better to have separately labeled vertical axes for these graphs and just remove the horizontal titles at the top.
15. Figure 4 (and probably the other figures): the figures seems to be low resolution, maybe a .jpg file? This leads to some distortion and blurring when zoomed in. I'd suggest using png or tif format to keep file sizes small and avoid image distortion.
16. Page 9, line 240: high HDR in the embryo seems to be ruled out experimentally, at least for all systems investigated thus far. Thus, the consideration of HDR in the embryo doesn't contribute much to the impact of the study and may be distracting from more important results.
17. Figure 5: I'm less happy with the model for this figure. It seems to me that the authors are not getting the timing right for the events they are trying to model and thus, they are not getting accurate results. This seems to partially stem from the authors decision to separate different aspects of resistance allele formation. This decision works fine for haploinsufficiency and for leaky expression, since these are not dependent on parental deposition. However, it starts getting less realistic afterward. I understand that this is just a model and that it's fine to investigate some variants that may not correspond to the reality of existing systems (since they are still plausible of possible future systems in different organisms or with different drive elements), but it may extend beyond plausibility on one case here. Below is my reasoning.

When looking at fitness effects (Figure 4), the authors can sort of make the case that they are examining times when parental deposition does not form a full early embryo resistance allele that is then present in all cells during most of development. These represent the “mosaic” individuals seen in *Drosophila* studies. Champer et al. 2019a even indicates that such individuals will almost always have normal drive conversion efficiency in the later meiotic phases, making it sensible to neglect this aspect. One could potentially even argue that in some species, the embryo may develop so quickly that there are many more “mosaic” individuals than those that get resistance alleles formed in the zygote or early embryo phases, making this a decent approximation, even

though the assumption is a little shaky. At any rate, even if it neglects the early embryo effects, the events modeled (reduced fitness) are still realistic for this class.

However, the next step in figure 5 (parental effects on gene transmission) seems to be a break from modeling things realistically. This is because it assumes formation of resistance alleles (or even drive conversion, which as noted above, should simply not happen at appreciable rates at this stage) that prevent germline conversion, but do not have other effects. However, if parental deposition is partially affecting germline cells, it is almost certainly affecting somatic cells and causing some variable reduction in fitness for females. Furthermore, as noted above in Champer et al. 2019a, “mosaic” individuals are usually not actually mosaic in the germline. The modeling assumes germline mosaicism but no somatic effects, which is not consistent with these findings.

Really, instead of restricting modeling to gene transmission, the author could simply keep the “mosaic” modeling in the “parental effects on fitness” section and change the “parental effects on gene transmission” to “early embryo resistance allele formation section”. This section could be placed before the “parental effects on fitness” section and deal with “complete” early embryo resistance allele formation. Modeling such effects would be straightforward. If parents have a drive allele, then simply change any wild-type alleles in the offspring to resistance alleles (or drive alleles if the authors want to model early embryo drive conversion, but this is not necessary). This would then be the offspring’s new genotype for all purposes. It would often have the effect of changing D/W individuals to D/R individuals, making them sterile if female (and of course preventing any drive conversion in the germline).

Right now, the drive conversion is prevented by the rate specified, but the females are still fully fertile. If resistance alleles form in the early embryo, the females would not be fertile. Thus, the effects would look more like a combination between Figures 4 and 5. For maternal deposition only, the load is around 0.93, varying only marginally as embryo resistance changes (so the drive actually performs better in terms of equilibrium load in this scenario). The main effect is slowing down the rate that the drive increases in frequency, which may be worth investigating in this or a future study.

18. In the discussion, perhaps the authors could cite the multiple gRNA bioRxiv manuscript (<https://www.biorxiv.org/content/10.1101/679902v1>) manuscript and discuss how their findings support its results for population suppression? The quest to eliminate functional resistance alleles with multiple gRNAs could result in situation where the drive loses efficiency and forms non-functional alleles, presenting a similar situation to that explored in the authors’ manuscript, thus raising its potential applicability to considerations in drive construction.

19. In the discussion or elsewhere might be useful to note that complete suppression is predicted to occur in panmictic models when the population growth rate at low density (always higher than 1 due to less competition) is less than $1/(1 - \text{load})$. Otherwise, a lower equilibrium population is reached. This could help people who are less familiar with modeling better understand the concept of load in the context of suppression.

Overall, the paper was an interesting read. I’m happy with it if the authors make an attempt to address some of the above points. I do particularly hope that the authors can either revise in response to point #17 or make a case that my reasoning is incorrect. If so, I can certainly recommend this article for publication in *Proceeding of the Royal Society B*.

Decision letter (RSPB-2019-1586.R0)

12-Aug-2019

Dear Dr Beaghton,

Your manuscript has now been peer reviewed and the reviews have been assessed by an Associate Editor. The reviewers' comments (not including confidential comments to the Editor) and the comments from the Associate Editor are included at the end of this email for your reference. As you will see, the reviewers' opinion of the manuscript is very positive, but they have raised some concerns which we would like to invite you to address in a revised version.

We do not allow multiple rounds of revision so we urge you to make every effort to fully address all of the comments at this stage. If deemed necessary by the Associate Editor, your manuscript will be sent back to one or more of the original reviewers for assessment. If the original reviewers are not available we may invite new reviewers.

Research ethics:

Use of animals and field studies:

It is a condition of publication that you make available the data and research materials supporting the results in the article. Datasets should be deposited in an appropriate publicly available repository and details of the associated accession number, link or DOI to the datasets must be included in the Data Accessibility section of the article

(<https://royalsociety.org/journals/ethics-policies/data-sharing-mining/>). Reference(s) to datasets should also be included in the reference list of the article with DOIs (where available).

Please submit a copy of your revised paper within three weeks. If we do not hear from you within this time your manuscript will be rejected. If you are unable to meet this deadline please let us know as soon as possible, as we may be able to grant a short extension.

Best wishes,
Professor Loeske Kruuk
Editor
mailto: proceedingsb@royalsociety.org

Reviewer(s)' Comments to Author:

Referee: 1

Comments to the Author(s)

In the manuscript, Beaghton and colleagues take the next step in studying the evolution of resistance to gene drive by exploring the influence of parental deposition and leaky expression of gene drive nucleases. They find that both can influence the equilibrium values of both driver

frequency and drive load. The paper is scientifically sound and mostly well written. I have several comments:

Lines 86-91 - I could use more of a preview in this paragraph. Instead of referring to "these factors", the authors could rehash what the factors are and give a bit more of a hint at the answer. It took me a while to understand exactly what the paper was about.

Line 107 (and throughout) - I struggled with mosaicism in the manuscript in general. It seems like leakiness and parental deposition would very likely lead to mosaic animals where fitness would depend (linearly, based on a threshold, etc.) on the proportion of cells that are WW/WD/WR/DD/DR/RR after editing. Perhaps the assumption is that all editing takes place early enough that mosaics are unlikely, but the discussion of somatic mutations would argue against that. Perhaps I'm just unclear on the assumption, but some clarification would help.

Line 107/114 - the use of the superscript 01/10/11 in two different contexts is confusing. Can one be changed?

Line 129 - how certain are you that for all parameter space, equilibrium is actually reached in 200 generations - that seems awfully short

"Parental effects on fitness" section - I had trouble following this section. It would benefit from some clarification about the question being asked and how it is addressed.

I would really like an intuitive explanation for why paternal deposition had a much stronger influence than maternal effects. It seems like they should be symmetrical. Can the authors provide any insight?

Lines 219-220 - The authors state that "However, there is no qualitative (and very little quantitative) change to the results if we instead assume that only W/D and W/R females are affected by the deposition." Is this worth putting in a supplemental table?

Mathematica - It might be useful to provide the Mathematica notebooks as a supplement or Figshare.

Referee: 2

Comments to the Author(s)

In this manuscript, the authors analyzed homing drives for population suppression using computational modeling. They specifically examined the effect of resistance allele formation in which the resistance alleles disrupt the target gene of the suppression drive (including alleles formed by parental deposition and in somatic cells by leaky nuclease expression). These alleles won't stop the drive like those that change the sequence but do not disrupt the target gene, but they can still have substantial effects on the outcome of the drive strategy. In particular, the authors find that such alleles can reduce the genetic load on the population, potentially preventing complete suppression of the population and allowing it to persist at a lower equilibrium population.

This is not a fundamentally new concept. Such resistance alleles have been included in a few previous models. Furthermore, in Deredec et al. 2008 and in a recent study posted to bioRxiv (<https://www.biorxiv.org/content/10.1101/679902v1>), it was even shown that complete population suppression would not occur for less efficient drives due to these same factors that the authors discuss. Nevertheless, the authors' treatment of this topic is quite focused (unlike the

bioRxiv study, which focuses on other matters and only briefly touches on the main points of this manuscript). It includes explicit treatment of leaky expression and parental deposition, and these topics are certainly an important enough that they deserve a thorough treatment. Indeed, the manuscript is particularly timely in light of the recent experimental publications of successful homing drives for population suppression. It is of overall high quality and should be quite suitable for publication with some modest revisions detailed below (sorry for the length - I get enthusiastic about gene drive). The only serious point is #17.

1. In the abstract, “non-functional” resistance alleles is used without much background. This could be very confusing to people outside the field of gene drive. The authors should consider describing how population suppression drives have a genetic target, and that non-function resistance alleles refer to the target gene. Same with paragraph two in the introduction section.
2. Later in the abstract, it is unclear why paternal nuclease deposition may be more serious than maternal deposition, particularly since maternal deposition would be expected to be far more common, all else being equal.
3. Page 2, line 55: The Oberhofer PNAS reference is not very good for noting progress with multiple gRNAs (also in the discussion section page 10 line 275). Their construct had far lower efficiency than one-gRNA drives in *Drosophila*. The other PNAS paper by Champer et al. in 2018 showed an improvement and should probably be reference here if the authors want to show progress with multiple gRNAs (not necessarily recommending a replacement of Oberhofer, just an addition). Perhaps also mention improved promoters for improving drive efficiency, such as Hammond 2018 on bioRxiv?
4. Page 3, line 68: “Leaky expression” was also observed in some of the *Drosophila* studies. It was noted in the Champer 2018 paper with the *vasa* promoter (in contrast to *nanos*, which didn’t have noticeable leaky expression) and in some of the Gantz/Bier papers with the *vasa* promoter.
5. Page 3, line 104: It’s a decent approximation to model parental effects to be the same with one or two drive alleles, but I don’t think it’s a good idea to model males as having a parental effect with the CRISPR system, since there is no hard evidence of this that I’m aware of. The authors are very aware of male parental deposition due to their previous experience with I-PpoI X-shredders, but a similar CRISPR X-shredder (Galizi et al., 2016) that they constructed did not show this phenomenon. Since the female gamete is much larger, it makes sense in general to assume that maternal deposition would usually have a much greater impact.

The reduced fertility they saw in the daughters with males of their homing suppression gene drives in *Anopheles* could probably be better explained by leaky expression of Cas9 rather than male deposition. This is quite clear in the *vasa* drive and the *nanos* drive from bioRxiv where the drive conversion efficiency of progeny with a drive mother is reduced. The lack of reduced drive conversion in the *zpg* lines indicates that reduced fitness is more likely due to leaky somatic expression. If there was substantial maternal or paternal deposition, then at least some resistance alleles would be formed in the early embryo and prevent drive conversion, which we do not see to a significant degree. The difference in egg count in females that had a drive mother or father could probably be better explained by imprinting or another phenomenon affecting the degree of leaky expression (especially since it seems different for the doublesex locus and the AGAP007280 gene in their bioRxiv paper).

While I believe that implementing this change would result in improvement in the accuracy/realism of the modeling, the ultimate qualitative outcomes would likely be similar whether maternal or both paternal and maternal nuclease deposition was modeled. Thus, if it

would not be possible to easily adjust the modeling, it should be sufficient to briefly note some of these considerations in the discussion of the manuscript.

6. Page 3 line 107 through end of paragraph: I'm a little unclear on one aspect of the model here. If there is parental nuclease deposition, it seems that two things both happen. A) W alleles may be converted to R or D alleles (usually R). B) All females with a drive parent will have a reduction in fitness, depending on which parents have drive alleles.

This seems to be pretty close to my understanding of how things would work, but it could perhaps get some clarification, since this section is pretty dense. Based on my understanding, maternal deposition would form resistance alleles (with no drive conversion) in a fraction of early embryos. This often makes the entire organism D/R or R/R, resulting in complete sterility for females. It would also prevent any drive conversion at all in the germline of males. Then, if there is not cleavage in the early embryo, there could be cleavage later in the embryo that results in a mosaic pattern of cleavage (seen in the *Drosophila* experiments and closely related to the early embryo cleavage rate - see supplemental material in Champer et al. 2019b). This could reduce the fitness of females by a varying amount and possibly affect drive conversion efficiency in the germline if this tissue is affected (this stage only seems to be what the authors model, rather than the "full" embryo resistance alleles formed before formation of mosaicism would take place). Up to this point, these effects could occur regardless of whether an individual actually received a drive allele. Finally, there is leaky somatic expression for individuals with a drive, which can occur whether the drive was received from a male or female parent. This will reduce the fitness of females, but probably not affect germline drive conversion efficiency. As in #5, the authors probably don't need to adjust their model if it is a good approximation, but they may want to note any differences in the above methods in their methods section if they agree with the above description of the mechanism. Either way, please try to improve the clarity of this section. The supplemental section helped, so one possible option is to reduce detail and reply more on the supplement.

7. Page 4, line 130-131. Using "e" for HDR (when e is the first letter in EJ) keeps throwing me off in this manuscript. Part of this I traced to these lines. I suggest flipping the order of and e in the parenthesis that is part of the phrase "We investigate variation in embryonic rates of homologous and end-joining repair (and e)". You could also flip the order before the parenthesis.

8. Figure 2 results: it might be interesting to point out earlier (it is discussed in Figure 3) that while the drive takes longer in the haploinsufficiency scenario, the final load is higher. This could actually be quite important (a 0.99 load is quite different from 0.95 in many scenarios). In this section, it may also be good to redefine load, which is current in the methods (which readers often skip).

9. Figure 3: the clarity of this figure would be increased if the vertical axis was changed from "allele frequencies and load" to "allele frequencies and load after 200 generations".

10. Figure 3: in (C) and (D), the rate of meiotic resistance is varied from the default value of 0.05. Since this plus the rate of HDR cannot exceed 1, and the default rate of HDR is 0.9, how are the two values related to each other? The rate of HDR would need to be lower than 0.9 if the rate of EJ repair is greater than 0.1 unless these events occur sequentially, but I didn't see any reference to this.

11. Figure 3: as the fitness costs and resistance rate increases, the rate at which the drive reaches a high load will be reduced. A good supplemental figure may show something relating to this (time to 90% of maximum genetic load?) if it could be done quickly/easily.

12. Page 8, line 219: the authors could explicitly say that W/W females with a drive parent would be very rare, which is why changing things for them has little effect on the overall outcome (even though they would undoubtedly be less affected).

13. Figure 4: Since the authors already do not distinguish between parents having one or two drive alleles, it's probably not necessary to separately model $w_{11} = w$ and $w_{11} = w^2$. Just keeping the former is fine with a note in the main text, and this could help make the graph a bit less crowded. This is low priority, though.

14. Figure 4: the vertical axis of this figure has both frequencies and load, but the graphs are one or the other. It may be better to have separately labeled vertical axes for these graphs and just remove the horizontal titles at the top.

15. Figure 4 (and probably the other figures): the figures seems to be low resolution, maybe a .jpg file? This leads to some distortion and blurring when zoomed in. I'd suggest using png or tif format to keep file sizes small and avoid image distortion.

16. Page 9, line 240: high HDR in the embryo seems to be ruled out experimentally, at least for all systems investigated thus far. Thus, the consideration of HDR in the embryo doesn't contribute much to the impact of the study and may be distracting from more important results.

17. Figure 5: I'm less happy with the model for this figure. It seems to me that the authors are not getting the timing right for the events they are trying to model and thus, they are not getting accurate results. This seems to partially stem from the authors decision to separate different aspects of resistance allele formation. This decision works fine for haploinsufficiency and for leaky expression, since these are not dependent on parental deposition. However, it starts getting less realistic afterward. I understand that this is just a model and that it's fine to investigate some variants that may not correspond to the reality of existing systems (since they are still plausible of possible future systems in different organisms or with different drive elements), but it may extend beyond plausibility on one case here. Below is my reasoning.

When looking at fitness effects (Figure 4), the authors can sort of make the case that they are examining times when parental deposition does not form a full early embryo resistance allele that is then present in all cells during most of development. These represent the "mosaic" individuals seen in *Drosophila* studies. Champer et al. 2019a even indicates that such individuals will almost always have normal drive conversion efficiency in the later meiotic phases, making it sensible to neglect this aspect. One could potentially even argue that in some species, the embryo may develop so quickly that there are many more "mosaic" individuals than those that get resistance alleles formed in the zygote or early embryo phases, making this a decent approximation, even though the assumption is a little shaky. At any rate, even if it neglects the early embryo effects, the events modeled (reduced fitness) are still realistic for this class.

However, the next step in figure 5 (parental effects on gene transmission) seems to be a break from modeling things realistically. This is because it assumes formation of resistance alleles (or even drive conversion, which as noted above, should simply not happen at appreciable rates at this stage) that prevent germline conversion, but do not have other effects. However, if parental deposition is partially affecting germline cells, it is almost certainly affecting somatic cells and causing some variable reduction in fitness for females. Furthermore, as noted above in Champer et al. 2019a, "mosaic" individual are usually not actually mosaic in the germline. The modeling assumes germline mosaicism but no somatic effects, which is not consistent with these findings.

Really, instead of restricting modeling to gene transmission, the author could simply keep the

“mosaic” modeling in the “parental effects on fitness” section and change the “parental effects on gene transmission” to “early embryo resistance allele formation section”. This section could be placed before the “parental effects on fitness” section and deal with “complete” early embryo resistance allele formation. Modeling such effects would be straightforward. If parents have a drive allele, then simply change any wild-type alleles in the offspring to resistance alleles (or drive alleles if the authors want to model early embryo drive conversion, but this is not necessary). This would then be the offspring’s new genotype for all purposes. It would often have the effect of changing D/W individuals to D/R individuals, making them sterile if female (and of course preventing any drive conversion in the germline).

Right now, the drive conversion is prevented by the rate specified, but the females are still fully fertile. If resistance alleles form in the early embryo, the females would not be fertile. Thus, the effects would look more like a combination between Figures 4 and 5. For maternal deposition only, the load is around 0.93, varying only marginally as embryo resistance changes (so the drive actually performs better in terms of equilibrium load in this scenario). The main effect is slowing down the rate that the drive increases in frequency, which may be worth investigating in this or a future study.

18. In the discussion, perhaps the authors could cite the multiple gRNA bioRxiv manuscript (<https://www.biorxiv.org/content/10.1101/679902v1>) manuscript and discuss how their findings support its results for population suppression? The quest to eliminate functional resistance alleles with multiple gRNAs could result in situation where the drive loses efficiency and forms non-functional alleles, presenting a similar situation to that explored in the authors’ manuscript, thus raising its potential applicability to considerations in drive construction.

19. In the discussion or elsewhere might be useful to note that complete suppression is predicted to occur in panmictic models when the population growth rate at low density (always higher than 1 due to less competition) is less than $1/(1 - \text{load})$. Otherwise, a lower equilibrium population is reached. This could help people who are less familiar with modeling better understand the concept of load in the context of suppression.

Overall, the paper was an interesting read. I’m happy with it if the authors make an attempt to address some of the above points. I do particularly hope that the authors can either revise in response to point #17 or make a case that my reasoning is incorrect. If so, I can certainly recommend this article for publication in Proceeding of the Royal Society B.

Author's Response to Decision Letter for (RSPB-2019-1586.R0)

See Appendix A.

Decision letter (RSPB-2019-1586.R1)

09-Oct-2019

Dear Dr Beaghton

I am pleased to inform you that your manuscript entitled "Gene drive for population genetic control: non-functional resistance and parental effects" has been accepted for publication in Proceedings B.

Open Access

Paper charges

Sincerely,

Professor Loeske Kruuk
Editor, Proceedings B
<mailto:proceedingsb@royalsociety.org>

Associate Editor:

Board Member

Comments to Author:

(There are no comments.)

Appendix A

Responses to Reviewers

Referee: 1

Comments to the Author(s)

In the manuscript, Beaghton and colleagues take the next step in studying the evolution of resistance to gene drive by exploring the influence of parental deposition and leaky expression of gene drive nucleases. They find that both can influence the equilibrium values of both driver frequency and drive load. The paper is scientifically sound and mostly well written.

Thanks to the reviewer for these comments. I have several comments:

Lines 86-91 - I could use more of a preview in this paragraph. Instead of referring to "these factors", the authors could rehash what the factors are and give a bit more of a hint at the answer. It took me a while to understand exactly what the paper was about.

To clarify, we propose replacing the last sentence (89-91) with:

If, as has been observed, nuclease expression causes strong fitness effects in drive heterozygotes, or if parental deposition of nuclease reduces offspring fitness or affects the genotype of their germline, we find that there can be reductions in the equilibrium frequency of the driving construct and the reproductive load imposed upon the population. Under such conditions, non-functional drive-resistant alleles can accumulate in the population and reduce the extent of population control.

Line 107 (and throughout) - I struggled with mosaicism in the manuscript in general. It seems like leakiness and parental deposition would very likely lead to mosaic animals where fitness would be depend (linearly, based on a threshold, etc.) on the proportion of cells that are WW/WD/WR/DD/DR/RR after editing. Perhaps the assumption is that all editing takes place early enough that mosaics are unlikely, but the discussion of somatic mutations would argue against that. Perhaps I'm just unclear on the assumption, but some clarification would help.

We do assume that mosaic individuals can be formed due to leakiness and parental deposition, and we explain our assumptions about mosaicism in Fig. 1 and in the Supplement; however, for greater clarity, we propose the following addition:

After line 102 and before line 104 (to clarify modelling of leakiness and parental effects), we added: Leaky expression of the nuclease in W/D females may lead to individuals that are mosaic in their soma, with a proportion of cells R/D due to end-joining repair (and D/D for homologous repair). We model this by a reduction in fitness of W/D heterozygote females. To model the action of deposited parental nuclease in the embryo, we assume that individuals may end up mosaic in their soma, affecting female fitness, and/or mosaic in their germline cells which alters gene transmission (Fig. 1).

Line 107/114 - the use of the superscript 01/10/11 in two different contexts is confusing. Can one be changed?

The superscripts 10,01,11 on various parameters are used to denote the same thing in all contexts: whether there is deposition from mother (10), father (01) or both (11). So, for example, w^{10} is for female fitness in an individual *with deposition from the mother (01)*; δ_e^{10} is the fraction of embryonic EJ repair in an individual *with deposition from the mother (01)*. Hopefully this is now clearer.

Line 129 - how certain are you that for all parameter space, equilibrium is actually reached in 200 generations - that seems awfully short. In general, we find that equilibrium is generally reached long before this time (see time dynamics in Fig. 2, and proposed additional Supplementary Fig. S1), and we have verified this for certain parameters by calculating up to 400 generations. However, we will make it clearer in the Figure and captions that the equilibrium results are after 200 generations.

"Parental effects on fitness" section - I had trouble following this section. It would benefit from some clarification about the question being asked and how it is addressed.

To make this clearer, we propose modifying lines 197-198 to:

It has been observed experimentally that the fitness of individuals can depend not only on their zygotic genotype, but also on the genotype of their parents, apparently due to parentally deposited nuclease creating mosaic offspring. Here, we investigate the consequences of fitness reductions due to parentally deposited nuclease for a strategy targeting a recessive female fertility gene, assuming there is neither partial haplo-insufficiency nor leaky expression. We investigate the sensitivity of the outcome (i.e., the allele frequency and load) to whether the fitness reductions in the offspring are caused by deposition from the mother only, the father only, or both, by varying the fitness parameters for each separately. We make no assumptions about whether maternal or paternal deposition is more likely – both have been observed for different constructs – and instead simply investigate the impact of deposition by either parent or both on the outcome.

I would really like an intuitive explanation for why paternal deposition had a much stronger influence than maternal effects. It seems like they should be symmetrical. Can the authors provide any insight?

Whether or not (for a given species or construct) it is shown experimentally that maternal vs paternal deposition has a stronger effect on the fitness (and/or germline transmission) *of the offspring*, here, we are investigating the sensitivity of the outcome of this gene drive strategy (targeting a female fertility gene) to whether the fitness reduction is from the mother, father or both. So, comparing the same amount of fitness reduction in the offspring from mothers only vs fathers only (or from both), we find that the outcome is more sensitive to the effect coming from a father (i.e., a stronger deleterious effect on the load and less population control). The explanation is that in a strategy targeting a female fertility gene, more zygotes get parental nuclease from the father than from the mother (i.e., more of them have D-bearing fathers than D-bearing mothers) because D/D and D/R genotypes are fertile if male and sterile if female.

We explain this in lines 205-207, but propose making it clearer by adding (in bold): The larger impact of paternal compared to maternal deposition **in a strategy targeting a female fertility gene** is because D/D and D/R genotypes are fertile if male and sterile if female, and therefore more zygotes have a D-bearing father than D-bearing mother.

We also propose adding this phrase to the abstract: We use population genetic modelling of a **strategy targeting a female fertility gene** to demonstrate that such alleles may be expected to accumulate, and thereby reduce the reproductive load on the population, if nuclease expression per se causes substantial heterozygote fitness effects or if parental (especially paternal) deposition of nuclease either reduces offspring fitness or affects the genotype of their germline.

Lines 219-220 - The authors state that "However, there is no qualitative (and very little quantitative) change to the results if we instead assume that only W/D and W/R females are affected by the deposition." Is this worth putting in a supplemental table?

The results for Fig. 4 were virtually indistinguishable, so instead of a supplement we propose to change the text here to: However, there is no qualitative (and **insignificant** quantitative) change to the results...

Mathematica - It might be useful to provide the Mathematica notebooks as a supplement or Figshare.

The calculations were carried out by using a 'subset' of parameters in a more general/complex gene drive programme that contains options for research not yet published (although we would determine how best to make the code available if requested by a reader).

Referee: 2

Comments to the Author(s)

In this manuscript, the authors analyzed homing drives for population suppression using computational modeling. They specifically examined the effect of resistance allele formation in which the resistance alleles disrupt the target gene of the suppression drive (including alleles formed by parental deposition and in somatic cells by leaky nuclease expression). These alleles won't stop the drive like those that change the sequence but do not disrupt the target gene, but they can still have substantial effects on the outcome of the drive strategy. In particular, the authors find that such alleles can reduce the genetic load on the population, potentially preventing complete suppression of the population and allowing it to persist at a lower equilibrium population.

This is not a fundamentally new concept. Such resistance alleles have been included in a few previous models. Furthermore, in Deredec et al. 2008 and in a recent study posted to bioRxiv (<https://www.biorxiv.org/content/10.1101/679902v1>), it was even shown that complete population suppression would not occur for less efficient drives due to these same factors that the authors discuss. Nevertheless, the authors' treatment of this topic is quite focused (unlike the bioRxiv study, which focuses on other matters and only briefly touches on the main points of this manuscript). It includes explicit treatment of leaky expression and parental deposition, and these topics are certainly an important enough that they deserve a thorough treatment. Indeed, the manuscript is particularly timely in light of the recent experimental publications of successful homing drives for population suppression. It is of overall high quality and should be quite suitable for publication with some modest revisions detailed below (sorry for the length - I get enthusiastic about gene drive). *The only serious point is #17.*

1. In the abstract, "non-functional" resistance alleles is used without much background. This could be very confusing to people outside the field of gene drive. The authors should consider describing how population suppression drives have a genetic target, and that non-function resistance alleles refer to the target gene. Same with paragraph two in the introduction section.

We propose changing the abstract to (bold): Even if these strategies are successful, it is almost inevitable that **non-functional resistant alleles will arise at the target site that are resistant to the drive but do not restore function**, and the impact of such sequences on the dynamics of control has been little studied.

We propose adding:

(55-57)...*"However, even if functional target site resistance is completely avoided, it is almost inevitable that non-functional resistant alleles **at the target site** will arise, at least at some frequency."*

2. Later in the abstract, it is unclear why paternal nuclease deposition may be more serious than maternal deposition, particularly since maternal deposition would be expected to be far more common, all else being equal.

Whether or not (for a given species or construct) it is shown experimentally that maternal vs paternal deposition has a more serious effect on the fitness (and/or germline transmission) of the offspring, here, we are investigating the sensitivity of the outcome of this gene drive strategy (targeting a female fertility gene) to whether the fitness reduction is from the mother, father or both. So, comparing the same amount of fitness reduction in the offspring from mothers only vs fathers only (or from both), we find that the outcome is more sensitive to the effect coming from a father (i.e., a stronger deleterious effect on the load and less population control) and relatively insensitive to effects coming from the mother. The explanation for this is that in a strategy targeting a female fertility gene, more zygotes get parental nuclease from the father than from the

mother (i.e., more of them have D-bearing fathers than D-bearing mothers) because D/D and D/R genotypes are fertile if male and sterile if female.

We explain this in lines 205-207, but propose making it clearer by adding (in bold): “The larger impact of paternal compared to maternal deposition **in a strategy targeting a female fertility gene** is because D/D and D/R genotypes are fertile if male and sterile if female, and therefore 207 more zygotes have a D-bearing father than D-bearing mother.”

We also propose adding this to the abstract: “We use population genetic modelling of a **strategy targeting a female fertility gene** to demonstrate that such alleles may be expected to accumulate, and thereby reduce the reproductive load on the population, if nuclease expression per se causes substantial heterozygote fitness effects or if parental (especially paternal) deposition of nuclease either reduces offspring fitness or affects the genotype of their germline.”

To make this clearer, we also propose modifying lines 197-198 to add further explanation: It has been observed experimentally that the fitness of individuals can depend not only on their zygotic genotype, but also on the genotype of their parents, apparently due to parentally deposited nuclease creating mosaic offspring. Here, we investigate the consequences of fitness reductions due to parentally deposited nuclease for a strategy targeting a recessive female fertility gene, assuming there is neither partial haplo-insufficiency nor leaky expression. We investigate the sensitivity of the outcome (i.e., the allele frequency and load) to whether the fitness reductions in the offspring are caused by deposition from the mother only, the father only, or both, by varying the fitness parameters for each separately. We make no assumptions about whether maternal or paternal deposition is more likely – both have been observed for different constructs – and instead simply investigate the impact of deposition by either parent or both on the outcome.

3. Page 2, line 55: The Oberhofer PNAS reference is not very good for noting progress with multiple gRNAs (also in the discussion section page 10 line 275). Their construct had far lower efficiency than one-gRNA drives in *Drosophila*. The other PNAS paper by Champer et al. in 2018 showed an improvement and should probably be reference here if the authors want to show progress with multiple gRNAs (not necessarily recommending a replacement of Oberhofer, just an addition). Perhaps also mention improved promoters for improving drive efficiency, such as Hammond 2018 on bioRxiv?

Added the Champer reference as suggested.

4. Page 3, line 68: “Leaky expression” was also observed in some of the *Drosophila* studies. It was noted in the Champer 2018 paper with the *vasa* promoter (in contrast to *nanos*, which didn’t have noticeable leaky expression) and in some of the Gantz/Bier papers with the *vasa* promoter.

Thanks for the suggestion, but these other studies do not have relevance to the suppression strategy mentioned here – we have changed ‘construct’ to ‘suppression drive’ to clarify this.

5. Page 3, line 104: It’s a decent approximation to model parental effects to be the same with one or two drive alleles, but I don’t think it’s a good idea to model males as having a parental effect with the CRISPR system, since there is no hard evidence of this that I’m aware of. The authors are very aware of male parental deposition due to their previous experience with I-PpoI X-shredders, but a similar CRISPR X-shredder (Galizi et al., 2016) that they constructed did not show this phenomenon.

Regarding no evidence of paternal effects using CRISPR, the reviewer is mistaken in this regard.

Whilst some of the CRISPR-based X-shredder strains showed no evidence of paternal effects, one strain tested in Galizi et al. 2016 showed a 50% reduction in hatching rate (Figure 2) that is the expected outcome of paternal deposition. These results are consistent with paternal deposition being a product of the nuclease (and its stability) and its integration site in the genome – something

we had already observed using the Ppo-I nuclease in Galizi et al. 2014. Additionally, these well documented locus-dependent differences in paternal deposition would also explain differences the reviewer alludes to between gene drives integrated at AGAP007280 vs AGAP004050 (doublesex).

Since the female gamete is much larger, it makes sense in general to assume that maternal deposition would usually have a much greater impact.

Actually, the difference between male and female gametogenesis is considerable, and cannot be distilled into differences in the size of the resulting gametes. Please note that in this study, we are not making any priori assumptions about whether maternal or paternal deposition causes more impact in the offspring (in our model these are parameters that can be varied) – rather we are investigating the impact of deposition by either parent or both on the outcome of the strategy, as mentioned above in response to point (2).

The reduced fertility they saw in the daughters with males of their homing suppression gene drives in *Anopheles* could probably be better explained by leaky expression of Cas9 rather than male deposition.

The fact that the fertility is higher in females with male drive parents (compared to those from female drive parents) is precisely why this is not explained by leaky expression, in which differences would not depend upon which parent carried the drive construct.

This is quite clear in the vasa drive and the nanos drive from bioRxiv where the drive conversion efficiency of progeny with a drive mother is reduced. The lack of reduced drive conversion in the zpg lines indicates that reduced fitness is more likely due to leaky somatic expression. If there was substantial maternal or paternal deposition, then at least some resistance alleles would be formed in the early embryo and prevent drive conversion, which we do not see to a significant degree. The difference in egg count in females that had a drive mother or father could probably be better explained by imprinting or another phenomenon affecting the degree of leaky expression (especially since it seems different for the doublesex locus and the AGAP007280 gene in their bioRxiv paper). We already have evidence of paternal nuclease deposition using CRISPR. It may be more pragmatic to consider this phenomenon than to invoke a hypothetical mechanism that is not currently supported by evidence. In any case, it seems the effect on the drive would be the same whether we called it paternal deposition or imprinting.

While I believe that implementing this change would result in improvement in the accuracy/realism of the modeling, the ultimate qualitative outcomes would likely be similar whether maternal or both paternal and maternal nuclease deposition was modeled. Thus, if it would not be possible to easily adjust the modeling, it should be sufficient to briefly note some of these considerations in the discussion of the manuscript.

As we have shown, the 'ultimate qualitative outcomes' are quite different for a female fertility strategy whether there is just maternal vs both paternal and maternal deposition, so it is important to consider them both as we have done here (i.e., effects of deposition from fathers as well as mothers).

6. Page 3 line 107 through end of paragraph: I'm a little unclear on one aspect of the model here. If there is parental nuclease deposition, it seems that two things both happen. A) W alleles may be converted to R or D alleles (usually R). B) All females with a drive parent will have a reduction in fitness, depending on which parents have drive alleles.

This seems to be pretty close to my understanding of how things would work, but it could perhaps get some clarification, since this section is pretty dense.

We propose adding in line 104 (to further clarify modelling of parental effects):

To model the action of deposited parental nuclease in the embryo, we assume that individuals may end up mosaic in their soma, affecting female fitness, and/or mosaic in their germline cells which alters gene transmission (Fig. 1).

Based on my understanding, maternal deposition would form resistance alleles (with no drive conversion) in a fraction of early embryos. This often makes the entire organism D/R or R/R, resulting in complete sterility for females. It would also prevent any drive conversion at all in the germline of males. Then, if there is not cleavage in the early embryo, there could be cleavage later in the embryo that results in a mosaic pattern of cleavage (seen in the *Drosophila* experiments and closely related to the early embryo cleavage rate - see supplemental material in Champer et al. 2019b). This could reduce the fitness of females by a varying amount and possibly affect drive conversion efficiency in the germline if this tissue is affected (this stage only seems to be what the authors model, rather than the “full” embryo resistance alleles formed before formation of mosaicism would take place). Up to this point, these effects could occur regardless of whether an individual actually received a drive allele. Finally, there is leaky somatic expression for individuals with a drive, which can occur whether the drive was received from a male or female parent. This will reduce the fitness of females, but probably not affect germline drive conversion efficiency. As in #5, the authors probably don't need to adjust their model if it is a good approximation, but they may want to note any differences in the above methods in their methods section if they agree with the above description of the mechanism. Either way, please try to improve the clarity of this section. The supplemental section helped, so one possible option is to reduce detail and reply more on the supplement.

The referee appears to want a model tightly constructed around a particular paper/set of experiments in *Drosophila* on biorxiv. We do not think this is the best approach – even other studies (that we cite) fall outside these narrow confines. Rather, we present a much more general modelling framework that allows for both paternal as well as maternal deposition and separates out the effects on gene transmission and fitness. We have added a new section on how these latter two effects may combine, which should help the referee's understanding.

7. Page 4, line 130-131. Using “e” for HDR (when e is the first letter in EJ) keeps throwing me off in this manuscript. Part of this I traced to these lines. I suggest flipping the order of δ and e in the parenthesis that is part of the phrase “We investigate variation in embryonic rates of homologous and end-joining repair (δ and e)”. You could also flip the order before the parenthesis. **Thanks, we have corrected this.**

8. Figure 2 results: it might be interesting to point out earlier (it is discussed in Figure 3) that while the drive takes longer in the haploinsufficiency scenario, the final load is higher. This could actually be quite important (a 0.99 load is quite different from 0.95 in many scenarios). In this section, it may also be good to redefine load, which is current in the methods (which readers often skip).

We propose adding:

(152-153) “To investigate the effect of leaky expression we compare the dynamics of allele frequencies and load (**i.e, the reduction in reproductive output by the population**) under three scenarios...”

Unless we have misunderstood the reviewer's question, we do point out here (157-158) that final load is higher, but drive takes longer, but propose adding:

The main outcome of the heterozygote fitness effect **due to partial haploinsufficiency** is to slow down the spread of the transgene and to increase the eventual equilibrium load.

9. Figure 3: the clarity of this figure would be increased if the vertical axis was changed from “allele frequencies and load” to “allele frequencies and load after 200 generations”. **Added this, and we will also make it clearer in other figure captions.**

10. Figure 3: in (C) and (D), the rate of meiotic resistance is varied from the default value of 0.05. Since this plus the rate of HDR cannot exceed 1, and the default rate of HDR is 0.9, how are the two values related to each other? The rate of HDR would need to be lower than 0.9 if the rate of EJ repair is greater than 0.1 unless these events occur sequentially, but I didn't see any reference to this.

Thank you for catching this – the results are correct, but the x-axis was mislabelled and has been corrected to range from 0 to 0.1.

11. Figure 3: as the fitness costs and resistance rate increases, the rate at which the drive reaches a high load will be reduced. A good supplemental figure may show something relating to this (time to 90% of maximum genetic load?) if it could be done quickly/easily. **Adding a plot of time to 90% could be problematic because the load does not always increase monotonically, so we propose adding a supplemental figure (Supp Fig. S1) that shows the full time dynamics for different fitness costs and rate of resistance...such a plot also shows that generally equilibrium is reached by 200 generations. We add a sentence referring to this to the end of lines (182-183): For both cases, the time taken for the load to reach its equilibrium value increases with higher fitness cost (Supplementary Fig. S1).**

12. Page 8, line 219: the authors could explicitly say that W/W females with a drive parent would be very rare, which is why changing things for them has little effect on the overall outcome (even though they would undoubtedly be less affected).

We have added: "However, there is no qualitative (and very little quantitative) change to the results if we instead assume that only W/D and W/R females are affected by the deposition, because W/W females with a drive parent are relatively rare."

13. Figure 4: Since the authors already do not distinguish between parents having one or two drive alleles, it's probably not necessary to separately model $w_{11} = w$ and $w_{11} = w^2$. Just keeping the former is fine with a note in the main text, and this could help make the graph a bit less crowded. This is low priority, though. **Perhaps not, but we would prefer to leave in just for completeness.**

14. Figure 4: the vertical axis of this figure has both frequencies and load, but the graphs are one or the other. It may be better to have separately labeled vertical axes for these graphs and just remove the horizontal titles at the top. **We have simply removed the side axes.**

15. Figure 4 (and probably the other figures): the figures seems to be low resolution, maybe a .jpg file? This leads to some distortion and blurring when zoomed in. I'd suggest using png or tif format to keep file sizes small and avoid image distortion. **We have ensured that the quality of the plots (which will be submitted separately) is high.**

16. Page 9, line 240: high HDR in the embryo seems to be ruled out experimentally, at least for all systems investigated thus far. Thus, the consideration of HDR in the embryo doesn't contribute much to the impact of the study and may be distracting from more important results.

Though we have not yet investigated this phenomenon experimentally, it seems perfectly plausible and there is no reason to think HDR would not occur in the embryo as it does in the germline later in development, so we think it is worthwhile to model. Indeed, early investigations had suggested that perhaps all homing is the result of deposited nuclease causing HDR in the early embryo. Champer et al. 2019a did not find evidence for homing being caused by deposited nuclease, but this does not mean the phenomenon could not occur in *Anopheles*, or if the gene drive were designed using a different set of germline promoters. We would hesitate to make the claim that a single absence of evidence is evidence of absence, and certainly would not treat it as a rule that applies to all gene drives across all insect species. This has already proven an incorrect line of reasoning if we compare

the differential effect of deposition on germline mosaicism between experiments in *Drosophila* (Champer et al. 2019a) vs *Anopheles* (Hammond et al. 2017), and when using different germline promoters (Hammond et al. 2018).

17. Figure 5: I'm less happy with the model for this figure. It seems to me that the authors are not getting the timing right for the events they are trying to model and thus, they are not getting accurate results.

The reviewer's main point here, in our understanding, is that experimentally, it is likely that parental fitness effects might accompany germline effects. Our model can treat effects on both fitness and gene transmission together simply by extending the parameter space considered, so we propose adding a new section called Combined Parental Effects (see below), and with a new figure (in the Supplement (Supp Fig. S2) due to length restrictions in the manuscript) showing the effects of varying fitness costs and EJ together if deposition is from the mother, father, or both, to address the reviewer's concerns in this regard:

Combined parental effects on fitness and gene transmission. Parental deposition can obviously affect both fitness and gene transmission, and while a full analysis of the joint effect is beyond the scope of this paper, representative results (for the case of embryonic cleavage always being repaired by end joining) are shown in Supplementary Fig. S2. As expected, increasing frequencies of deposition-associated embryonic cleavage lead to reductions in the equilibrium frequency of the D allele and the equilibrium load in this case too.

This seems to partially stem from the authors decision to separate different aspects of resistance allele formation.

This decision works fine for haploinsufficiency and for leaky expression, since these are not dependent on parental deposition. However, it starts getting less realistic afterward. I understand that this is just a model and that it's fine to investigate some variants that may not correspond to the reality of existing systems (since they are still plausible of possible future systems in different organisms or with different drive elements), but it may extend beyond plausibility on one case here. Below is my reasoning.

When looking at fitness effects (Figure 4), the authors can sort of make the case that they are examining times when parental deposition does not form a full early embryo resistance allele that is then present in all cells during most of development. These represent the "mosaic" individuals seen in *Drosophila* studies. Champer et al. 2019a even indicates that such individuals will almost always have normal drive conversion efficiency in the later meiotic phases, making it sensible to neglect this aspect. One could potentially even argue that in some species, the embryo may develop so quickly that there are many more "mosaic" individuals than those that get resistance alleles formed in the zygote or early embryo phases, making this a decent approximation, even though the assumption is a little shaky. At any rate, even if it neglects the early embryo effects, the events modeled (reduced fitness) are still realistic for this class.

However, the next step in figure 5 (parental effects on gene transmission) seems to be a break from modeling things realistically. This is because it assumes formation of resistance alleles (or even drive conversion, which as noted above, should simply not happen at appreciable rates at this stage) that prevent germline conversion, but do not have other effects.

As mentioned above, the model *does* potentially include both effects together (and indeed, leaky expression could also additionally be combined with parental effects at the same time if desired, by entering the appropriate parameters into our model). However, we wanted to evaluate the impacts on the strategy of fitness reduction or germline transmission due to parental deposition separately, to understand what impact each has in isolation. However, we completely agree that it is important

to also include plots showing the combined effects of gene transmission and fitness since this may be a likely scenario, and we propose a new section and plot S2 as we suggest above.

Also in response to the reviewer's comment that: "drive conversion, which as noted above, should simply not happen at appreciable rates at this stage ", we have seen many instances in which maternally deposited nuclease results in germline mosaicism that affects the rate of homing in the progeny. For example, Figure 3c in Hammond et al. 2017 shows very clearly a near complete loss of homing in males that had a maternal, but not paternal, dose of the nuclease. In Hammond et al. 2018, we observed a similar maternal effect on homing when using the nanos promoter (both males and females that inherited the gene drive from their mother showed lower homing than those inheriting the drive from their father). Given the different activities of these promoters in different organisms, including other mosquito species, it is perhaps difficult to draw direct comparison between similar investigations in other insects – such as those of Champer et al 2019a.

However, if parental deposition is partially affecting germline cells, it is almost certainly affecting somatic cells and causing some variable reduction in fitness for females. Furthermore, as noted above in Champer et al. 2019a, "mosaic" individuals are usually not actually mosaic in the germline. The modeling assumes germline mosaicism but no somatic effects, which is not consistent with these findings.

Again, we note that the model can treat both germline mosaicism and somatic effects at the same time; we chose parameters in order to evaluate the effects of each separately, but now are also including results where the two effects (germline and somatic mosaicism) are combined (new section).

Just to respond in more detail to the reviewer's statement that if cells are affected in one area they must be in another, in Champer et al. 2019a there were mosaic individuals (presumably somatic?) that did not show mosaicism of the germline (in contrast to our own data, as stated above)), supporting the hypothesis that deposition-induced mosaicism can differentially affect somatic and germline tissues. We don't think that this evidence conflicts with our assumptions.

Really, instead of restricting modeling to gene transmission, the author could simply keep the "mosaic" modeling in the "parental effects on fitness" section and change the "parental effects on gene transmission" to "early embryo resistance allele formation section". This section could be placed before the "parental effects on fitness" section and deal with "complete" early embryo resistance allele formation. Modeling such effects would be straightforward. If parents have a drive allele, then simply change any wild-type alleles in the offspring to resistance alleles (or drive alleles if the authors want to model early embryo drive conversion, but this is not necessary). This would then be the offspring's new genotype for all purposes. It would often have the effect of changing D/W individuals to D/R individuals, making them sterile if female (and of course preventing any drive conversion in the germline).

Thanks for the suggestion, but due to repeated experimental observations of mosaicism, we think that our mosaic model may overall be more realistic rather than a model only featuring "complete" conversion (Hammond 2017, Papathanos); so instead of adding a section focussing on 'early resistance formation' that does not include mosaics and reverts to full-conversion only, we feel that the new section that we propose - showing the combined effects of fitness and germline transmission in our mosaic model – is the best way to address the reviewer's concerns.

Right now, the drive conversion is prevented by the rate specified, but the females are still fully fertile. If resistance alleles form in the early embryo, the females would not be fertile. Thus, the effects would look more like a combination between Figures 4 and 5.

See new Supp Fig. 2 in the new proposed section, which combines reductions in fertility and gene transmission.

For maternal deposition only, the load is around 0.93, varying only marginally as embryo resistance changes (so the drive actually performs better in terms of equilibrium load in this scenario). The main effect is slowing down the rate that the drive increases in frequency, which may be worth investigating in this or a future study. We are not clear which plot the reviewer is referring to, and assume it is maternal deposition in Fig. 5 c& d? We find that there is not much effect on the dynamics, see plots below with different colour lines for varying embryonic rate of EJ + HR ($\delta_e + e_e$) while keeping EJ/HDR ratio at the meiotic value:

18. In the discussion, perhaps the authors could cite the multiple gRNA bioRxiv manuscript (<https://www.biorxiv.org/content/10.1101/679902v1>) manuscript and discuss how their findings support its results for population suppression? The quest to eliminate functional resistance alleles with multiple gRNAs could result in situation where the drive loses efficiency and forms non-functional alleles, presenting a similar situation to that explored in the authors' manuscript, thus raising its potential applicability to considerations in drive construction. ? We generally prefer not to site/discuss unpublished papers, unless we need to refer to a specific experimental observation.

19. In the discussion or elsewhere might be useful to note that complete suppression is predicted to occur in panmictic models when the population growth rate at low density (always higher than 1 due to less competition) is less than $1/(1 - \text{load})$. Otherwise, a lower equilibrium population is reached. This could help people who are less familiar with modeling better understand the concept of load in the context of suppression. This has now been added in the Supplement (as the manuscript is already close to the length restriction).

Overall, the paper was an interesting read. I'm happy with it if the authors make an attempt to address some of the above points. I do particularly hope that the authors can either revise in response to point #17 or make a case that my reasoning is incorrect. If so, I can certainly recommend this article for publication in Proceeding of the Royal Society B.

Thanks very much for the helpful suggestions, most of which we have incorporated into the new version; and hopefully we have satisfactorily addressed the reviewer's most important concern in point #17 about fitness effects accompanying germline effects due to parental deposition by including a new section and a plot (S2) that shows the relevant results.